# Simple Masked Training Strategies Yield Control Policies That Are Robust to Sensor Failure

**Skand Peri, Bikram Pandit, Chanho Kim, Li Fuxin, Stefan Lee**
Oregon State University
**Project Page:** https://pvskand.github.io/projects/RME

**Abstract:** Sensor failure is common when robots are deployed in the real world, as sensors naturally wear out over time. Such failures can lead to catastrophic outcomes, including damage to the robot from unexpected robot behaviors such as falling during walking. Previous work attempted to address this problem by recovering missing sensor values from the history of states or by adapting learned control policies to handle corrupted sensors through fine-tuning during deployment. In this work, we propose training reinforcement learning (RL) policies that are robust to sensory failures. We use a multimodal encoder designed to account for these failures and a training strategy that randomly drops a subset of sensor modalities, similar to missing observations caused by failed sensors. We conduct evaluations across multiple tasks (bipedal locomotion and robotic manipulation) with varying robot embodiments in both simulation and the real world to demonstrate the effectiveness of our approach. Our results show that the proposed method produces robust RL policies that handle failures in both low-dimensional proprioceptive and high-dimensional visual modalities without a significant increase in training time or decrease in sample efficiency, making it a promising solution for learning RL policies that are robust to sensory failures.

**Keywords:** Reinforcement Learning, Robustness, Sensorimotor Learning

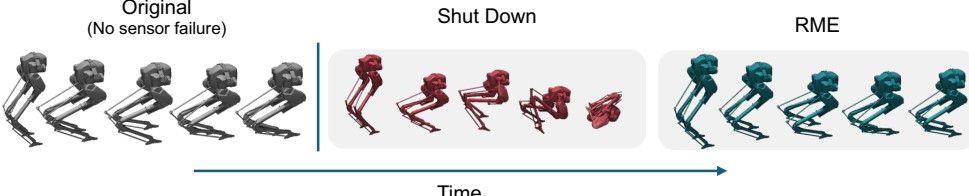

Figure 1: *Motivating Example*: When the `left-shin` sensor of the bipedal robot Cassie fails, a simple *shut down* operation leads it to topple down (**middle**) as opposed to our RME trained policy (**right**) which continues to walk similarly to the scenario with no sensory failure (**left**).

## 1 Introduction

Sensory failure in robots is both pervasive and unavoidable. What should a robot do next when it has detected a sensory failure or corruption? A *shut down* action is one option that an agent could afford; however, doing so safely for dynamical systems with complex robot morphology in uncertain terrains is a challenging control task in its own right. A straightforward option that cuts power to the motors could result in hazardous behavior, causing harm to the robot itself and other entities in the environment. For instance, the bipedal robot shown in a Figure 1 (middle) immediately falls down when power is cut during a squatting motion.

In such a scenario where the robot detects a sensory failure, one would like the robot to function to the best of its abilities and not collapse. For instance a robot failing in a warehouse would

8th Conference on Robot Learning (CoRL 2024), Munich, Germany.

cause disruptions to the operations. Additionally, consequences of such failures can be exacerbated for robots deployed for critical rescue operations [1] or in scenarios where immediate repair is unavailable [2]. Hence, it is crucial to have robots that are robust to sensor failures to help mitigate the risk of catastrophic outcomes and provide best-effort control until repairs can be made.

Several works have attempted to address the lack of sensory robustness from various perspectives. From a classical control viewpoint, robust model predictive control (MPC) has been thoroughly examined [3, 4]. For example, a common approach to managing sensor corruptions as demonstrated in [5], involves incorporating an additional feedback controller to address the deviations caused by the corruption. On the learning end, [6, 7, 8] adopt a continual learning setup where the control policy of the robot is adapted explicitly to the missing sensor during deployment, and [9, 10, 11] impute the missing values of the sensors from the history of states. The above proposed works try to address this issue after the control policy is trained by either adapting the policy or training another model to correct the values of the missing sensors (imputation networks).

Instead of addressing this issue in a post hoc manner, this work aims to train reinforcement learning (RL) control policies that are robust to sensory failures, enabling the robot to function *without* further fine-tuning or adaptation during deployment. As an example, we demonstrate this phenomenon in an experiment with Cassie the bipedal robot for the task of blind walking in Figure 1. When a standard walking policy trained with RL [12] is deployed on the robot with a broken `left-shin`, a native *shut down* operation leads to Cassie falling down (middle). However, when the policy takes into account such a failure we barely observe a difference in gaits of the walking policy (right). In this work, we focus on the setting where sensor failures are presumed to already be detected and develop best-effort policies that are robust to this loss of information.

To achieve this, we propose a straightforward multimodal encoder and incorporate a simple modality dropout schema where we randomly drop a subset of sensors during training so that the resulting policy is robust to missing sensors during test time. Further, we also extend this idea to multiple high-dimensional modalities such as multiview RGB and depth images in robotic manipulation and demonstrate its effectiveness for model-based and model-free reinforcement learners. In contrast to prior work on learning masked multimodal visual features in model-based RL [13, 14], our approach includes both low- and high-dimensional sensor modalities, does not require modality reconstruction, and results in a policy that can operate on arbitrary subsets of modalities during deployment.

Our experiments demonstrate that the proposed approach is effective across multiple robot embodiments, diverse tasks (tabletop pick & place, mobile manipulation, and bipedal locomotion), and even in sim2real deployment of a highly-dynamic bipedal robot. For most tasks, our approach incurs only a minimal increase of wall-clock training time and decrease in sample-efficiency – making it an easy-to-implement, efficient, and effective strategy to produce policies robust to sensor failure.

## 2   Related Works

**Sensor value imputation.** One common technique to deal with missing modalities is to predict them. This is a well studied problem in databases [9], machine learning [15, 16, 17, 18, 19] and model predictive control (MPC) [5, 3, 4]. In RL from high-dimensional sensory inputs such as RGB images, InformedDreamer [11] proposed to predict privilege modalities from latent state – however, one major drawback of this approach as noted in [20], is that not all privilege information is predictable from a set of input modalities. On similar lines [21] also proposed to predict linear and angular velocity for autonomous vehicles in the context of off-road robot driving. They also observe that predicting high-dimensional modalities in multi-modal world models can be difficult and lead to blurry outputs in RGB images and lidar maps.

**Handling missing modalities:** Another line of works attempted to learn to generate state representations robust to missing modalities without recovering them explicitly. For instance, [22] introduces CCM to address the sensor corruption problem in RL for images and force sensors. CCM trains an multimodal encoder that is flexible with respect to missing modalities by representing a latent state distribution as a product of per-modality distributions. However, proprioceptive

observations are never corrupted in [22] and their experiments were limited to a single peg-in-hole setting with only single-modality failures.

[23] proposed a Sensor Dropout (SD) strategy for training an RL agent for autonomous driving, in which, instead of randomly dropping neurons as in standard dropout, they randomly select modalities and zero out their features. They employ an MLP to fuse the modality features. While an MLP is a suitable architecture for lower numbers of modalities, we find that its performance can suffer for higher numbers of modalities, as shown in our experiments in Section 8.11. Additionally, this method was demonstrated with only three modalities on a single task with a 2-d action space. In contrast, we perform a wide range of experiments ranging from four high dimensional modalities (images and depth maps) and up to seven low-dimensional sensors with multiple complex action spaces (6-d for Walker, 10-d for bi-pedal walking, 7-d for manipulation, 13-d for loco-manipulation tasks).

**Multimodal policies.** Another way to address the availability of multiple sensors is using mutlimodal policies – policies trained on multiple inputs. Different observations may provide complementary information that is helpful for the agent to achieve its task – for example robotic manipulators are more sample-efficient when provided with first-person RGB inputs via a wrist camera [24]. Cross-modal policies that operate on multiple modalities [25, 26, 27] have been successful in achieving fine grained manipulation, but they end up fusing the representations from each modality, making it infeasible to function when a subset of modalities is missing.

RPT [28] has investigated learning pre-trained representations for robot learning from a dataset of trajectories. In particular, they employ a masking mechanism where a subset of modalities are masked out and are predicted from the rest. However, this work differs from ours in two aspects: (a) their primary goal was to learn transferable pretrained representations for robotic tasks, and (b) RPT includes losses that encourage explicit reconstruction of missing modalities which is shown to be detrimental to performance [20] and we find the same. MUTEX [29] uses multiple modalities like text instructions, video demos, speech and image goals along with robot states to train a policy that is robust to missing modalities. However, all multi-modal transformer based models such as MUTEX are limited to imitation learning where access to high fidelity video demonstrations are assumed. Our work focuses primarily on reinforcement learning policies where the agent has to learn the behavior from trial-and-error where transformer based encoders have been empirically shown to be unstable [30]. For more related works, please refer to Section 8.1.

## 3    Preliminaries

We pose our problem as an infinite-horizon Partially Observable Markov Decision Process (POMDP) [31] defined $(\mathcal{S}, \mathcal{A}, \mathcal{T}, \mathcal{R}, \mathcal{O})$. $\mathcal{S}$ represents the complete state space, which is usually not accessible to the robot. The observation space, $\mathcal{O} = \mathcal{O}^1 \times \mathcal{O}^2 ... \times \mathcal{O}^m$ is a set of $m$ sensory observations or modalities[1] such as proprioceptive state, multi-view RGB or depth images. $\mathcal{A} \in \mathbb{R}^d$ is a $d$-dimensional continuous action space, $\mathcal{T} : \mathcal{O} \times \mathcal{A} \to \mathcal{O}$ is the transition function, and $\mathcal{R} : \mathcal{O} \to \mathbb{R}$ is the reward function. The goal of the agent is to learn a policy $\pi : \mathcal{O} \to \mathcal{A}$ that maximizes the expected sum of discounted rewards; $\max_\pi \mathbb{E}_\pi [\sum_{t=1}^{\infty} \gamma^t \mathcal{R}(s_t)]$, where $\gamma \in [0, 1)$ is the discount factor.

While in principle, our proposed encoder design can be applied to any model-based or model-free method, we show experiments on model-based TD-MPC2 [32] and model-free PPO [33] agents.

**TD-MPC2.** TD-MPC2 has shown strong results across diverse robotic environments. Like many other model-based RL algorithms, it consists of (a) world model training and (b) policy learning.

World model learning. While most existing model-based methods such as Dreamer [34, 35, 36] learn a generative model of the world through supervision that includes a state reconstruction loss, TD-MPC2 learns an transition dynamic using supervision from rewards and TD-learning [37] alone.

---

[1]In this work, we interchangeably use sensors and modalities.

Specifically, the TD-MPC2 world model consists of the following components:

$$
\begin{aligned}
\text{Representation:} & \quad z_t = h_\phi(o_t, o_{t-1}, o_{t-2}) \\
\text{Dynamics:} & \quad \hat{z}_t = f_\phi(z_{t-1}, a_{t-1}) \\
\text{Reward:} & \quad \hat{r}_t = R_\phi(z_t) \\
\text{Value:} & \quad \hat{q}_t = Q_\phi(z_t, a_t) \\
\text{Policy prior:} & \quad \hat{a}_t = \pi_p(z_t)
\end{aligned}
\tag{1}
$$

where $o$ and $a$ are the observations and actions and $z$ is the learned latent which is further used to obtain a policy. Denoting the cross-entropy loss as CE and the `stop-gradient` operator as sg, the world model is trained using the following objective:

$$
\mathcal{L}(\phi) = \mathbb{E}_{(s,a,r,s')_{0:T} \sim \mathcal{B}} \left[ \sum_{t=0}^{T} \lambda^t \big( \underbrace{\|\hat{z}_t - \text{sg}(h_\phi(o_t, o_{t-1}, o_{t-2}))\|_2^2}_{\substack{\text{latent} \\ \text{consistency}}} + \underbrace{\text{CE}(\hat{r}_t, r_t)}_{\substack{\text{reward} \\ \text{prediction}}} + \underbrace{\text{CE}(\hat{q}_t, q_t)}_{\substack{\text{value} \\ \text{prediction}}} \big) \right]
\tag{2}
$$

where the expectation is over samples from a reply buffer $\mathcal{B}$ collected under $\pi_p$. The latent consistency term ensures the predicted latent $\hat{z}_t$ is similar to the embedding obtained by the representation network $h_\phi$ for the real observation at time $t$. The remaining terms encourage the predicted reward $\hat{r}_t$ and value $\hat{q}_t$ match a discrete regression encoding [38, 36] of the true reward $r_t$ and TD-estimated value $q_t = r_t + \bar{Q}(\hat{z}_t, p(\hat{z}_t))$, where $\bar{Q}$ is the exponential moving average of $Q_\phi$ [39].

Policy learning. TD-MPC2 trains the policy prior $p$ with the maximum entropy RL [40] objective:

$$
\mathcal{L}_p(\phi) = \mathbb{E}_{(s,a)_{0:T} \sim \mathcal{B}} \left[ \sum_{t=0}^{H} \lambda^t \left[ \alpha Q(z_t, \pi_p(z_t)) - \beta \mathcal{H}(\pi_p(a_t \mid z_t)) \right] \right]
\tag{3}
$$

where $z_t$ is given by equation (1). While training with the objective $\mathcal{L}_p$, only network parameters of $\pi_p$ are updated. TD-MPC2 employs gradient-free model predictive control via Model Predictive Path Intergral (MPPI) [41] that samples actions using rollouts from the learned world model. For more details, we refer the reader to the TD-MPC2 paper [32].

**PPO.** For model-free agents trained with PPO [33], a policy network $\pi(z_t)$ is optimized by using a clipped objective function to limit the magnitude of policy updates to maximize the expected discounted return directly, which is predicted by a learned critic $c(z_t)$ that is trained to estimate the value function simultaneously.

## 4 Learning Policies Robust to Sensor Failure

To support learning policies that perform well in the face of sensor failures, we focus on designing representation encoder networks that are flexible to the number of input modalities and corresponding modality dropout training strategies that encourage strong policy performance even when individual modalities are unavailable. Our resulting **Robust Multi modal Encoder** (RME) is simple to integrate with existing model-based and model-free algorithms, accommodates failures in both high-level (visual) and low-level (proprioceptive) sensors, and demonstrates similar (or better) sample-efficiency and wall-clock training time compared to single-modality baselines. The design of RME is straightforward and shows strong robustness to sensor failures when paired with modality dropout during training – in both simulated and real robotic deployments.

**Robust Multimodal Encoder** (RME). Our RME model consists of a multimodal transformer and individual modality encoders (as shown in Figure 2).

Multimodal Transformer. A common choice when combining multiple modalities in other domains is to use transformers [42] as used in [43, 44, 45]. Though widely used as sequence models, transformers without positional encoders operate as order-invariant set functions and offer a flexible way to retain or remove modalities simply by excluding them from the input set. Considering $M$ modalities $o^{(1)}, ..., o^{(M)}$ and corresponding modality encoders $f^{(1)}, ..., f^{(M)}$ which map each modality to a $d$-dimensional observation encoding, we can write the multimodal transformer as the function TF,

$$
h^{(1)}, ..., h^{(M)}, h^{(\text{CLS})} = \text{TF}\left( f^{(1)}(o^{(1)}) + e^{(1)}, ..., f^{(m)}(o^{(m)}) + e^{(m)}, e^{(\text{CLS})} \right),
\tag{4}
$$

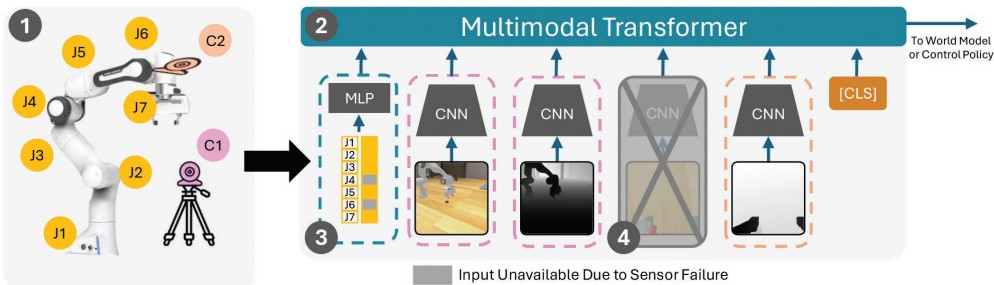

Figure 2: **Robust Multimodal Encoder** (RME): We propose a *simple*, yet effective modification to learn robust representations within existing RL frameworks. ❶ We consider each robot sensor as a modality that can be potentially missing during deployment. ❷ The proposed RME architecture consists of both low-dimensional proprioceptive inputs as well as high dimensional image inputs that are encoded using appropriate encoders (MLP, CNNs) and fed into a transformer. We employ different dropout mechanisms for low and high dimensional modalities. ❸ Given the proprioceptive (or any other low-dimensional) input, we append a mask vector where 1/0 indicate the presence/absence of the sensor respectively. ❹ For high-dimensional modalities, we simply drop the modality shaded in gray while training.

where $e^{(m)}$ is a $d$-dimensional learned modality type embedding akin to those used in multimodal models [46] and $e^{(\texttt{CLS})}$ is a learned input that is always present to aggregate information across modalities [47]. When using the RME in model-based or model-free methods, we use the output vector $h^{(\texttt{CLS})}$ as the representation encoding of the input modalities. In practice, individual modalities might be represented via multiple vectors, for instance representing visual modalities via many patches as done in vision transformer-based encodings [48, 49]. However, in this work, we consider only the single vector case to reduce computational costs from processing long sequences with transformers and to provide a uniform interface for all considered sensor modalities.

We note that the transformer-based image autoencoder from [14] has a similar structure for multi-view visual representation learning in MBRL – taking patch-based encodings of either single- or multi-view observations. However, [14] trains this model independently and then uses the feature encoder as representation for separate multi- or single-view policies; not producing a policy that is capable of adapting to sensor failure at runtime. In practice, many combinations of failed sensors are possible and learning independent policies for each is a costly solution.

Encoding Visual Sensors. For high-dimensional sensors such as RGB or depth images, we can accommodate any encoder $f$ that summarizes the image as a vector. For ease of comparison, we retain the image encoding network from TD-MPC2 [32] consisting of four conv layers with ReLU activations. The final spatial feature volume is flattened into a 512-dim vector. Each visual modality instantiates a different parameterization of this architecture.

Accommodating Low-Dimensional Sensors. Proprioceptive states $o^{(p)}$ are typically comprised of a combination of $K$ low-dimensional sensors such as IMUs and joint encoders where each sensor $o_k^{(p)}; k \in K$ is a $\{2\text{-}7\}$–dimensional real valued vector. In early experiments, we found treating each sensor as an independent input to the multimodal transformer (analogous to the visual features) was highly unstable for policy training. To overcome this while still allowing for individual sensors to fail, we adopt a different strategy for proprioception inputs. Specifically, for every sensor $o_k^{(p)}$, we have a corresponding binary mask $\mathcal{M}_k^{(p)}$ which is set to 0 if a sensor is dropped out, or otherwise to 1. These masks are concatenated $[o^{(p)}, \mathcal{M}_1^{(p)}, \mathcal{M}_2^{(p)}, ..., \mathcal{M}_K^{(p)}]$, that is fed to a single modality encoder ($f^{(\texttt{MLP})}$) to obtain our representation for proprioception.

Modality Dropout. When sampling a training batch, we randomly drop a subset of the $M$ sensor modalities. Specifically, we select a random subset of modalities with random size 1 to M-1. We then drop out sensors from the subset individually with probability $p$. This scheme ensures that at

least one modality always remains. Implicit in this method is the assumption that the remaining modalities are sufficient for the policy to reason about correct actions.

## 5 Experiments

The following experiments study three key questions: How robust is RME to proprioceptive sensory failures for locomotion tasks (Sec. 5.1)? Can RME scale with multiple high-dimensional modalities such as RGB & depth images and maintain its robustness to sensory failures during deployment (Sec. 5.2)? And can a control policy trained with RME transfer to the real world (Sec. 5.3)?

### 5.1 Sensor failures in robotic proprioceptive states

**Experimental setting.** We first consider tasks where individual sensors in the proprioceptive states of the robot encounter a sensory failure. We consider two different embodiments shown to the left of Figure 3 – Walker from the DMControl suite [50] and a bipedal robot, Cassie. For Walker, we show results on four different tasks {Walk, Run, Walk-back and Run-back} and for Cassie, we show results on walking. For walker, we consider 7 sensors – {torso, thigh (r/l), leg (r/l) and foot (r/l)} where l and r indicate left or right. For Cassie, we consider 4 sensors – {tarsus (l/r) and shin(l/r) encoders}. In all our experiments, we do not drop any sensors that are part of the action space or are required by the underlying controller – in other words, the sensors that are dropped are non-actuated sensors. Baseline Walker policies are trained with TD-MPC2 whereas Cassie uses the model-free PPO algorithm [33] as in prior work [51]. For each, we present results when replacing the representation encoder with our RME model and training with modality dropout. For baseline policies, we set failed sensors to values of 0 during inference – essentially performing a fixed in-distribution imputation. We also show results from an imputation baseline where failed sensor values are predicted from a history of 10 previous states using a trained 3 layer MLP with LayerNorm [52] and Mish activation [53]. The final layer is a linear transformation without any activations. For fairness, the imputation network is trained on transitions from the replay buffer collected during RME+{TD-MPC2 / PPO} training. For Walker, we consider 200k transitions and for Cassie we train the imputation network on 1B transitions in simulation.

**Results.** We consider single sensor failures and report the sensor failure that causes the minimum (largest impact) and maximum (smallest impact) return for the task as well as the mean return over sensor failure conditions. For context, baseline policies without sensor failures achieve returns of 900-950 for Walker and 320-350 for Cassie. From results in Figure 3 *(left)*, we observe that RME maintains comparably high average returns across different tasks and sensor failures. The simple imputation by zeros strategy sometimes leads to performant policies when the failed sensor does not seem relevant to the learned policy (e.g., left-foot in Walker (Walk) max), but is catastrophic in the worst cases (e.g., min returns 5-20x lower than max for Walker). Using learned imputation narrows this variance while raising mean performance – achieving lower max and higher min returns – but still underperforms. For the more complex Cassie robot, zeros imputation fails to produce a viable policy entirely and learned imputation suffers significant reductions as well compared to RME. We hypothesize that errors in learned imputation may have compounding effects if predicted sensor values drive the policy to out-of-distribution states where the imputation network is less well trained.

### 5.2 Sensor failures in high-dimensional modalities

**Experimental setting.** We also consider sensor failures in the context of robotic manipulation for high dimensional modalities such as images. Specifically, we conduct robotic manipulation experiments in simulation on the Maniskill2 [54] benchmark and consider first- and third person RGB and depth images as additional modalities. We consider three pick & place tasks (PickCube, StackCube, PickYCB) and two mobile manipulation tasks (OpenDrawer, OpenCabinet).

We consider both *Easy* settings where only single modalities (e.g., first-person depth) are dropped and *Hard* settings where *all but one* modalities are dropped. We report results for *Easy* in the

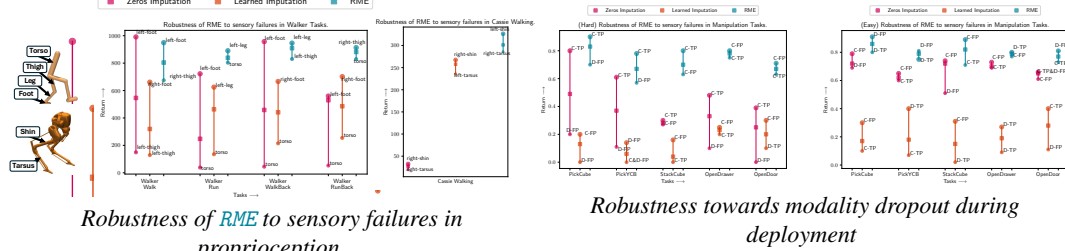

*Robustness of RME to sensory failures in proprioception*

*Robustness towards modality dropout during deployment*

Figure 3: **(left)** *Robustness of RME to sensory failures in proprioception:* We compare policies trained via proprioceptive masking mechanism ❸ against imputation baselines. We find that RME based policy is significantly robust to sensory failures. **(right)** *Robustness towards modality dropout during deployment* (Hard:) We test MMWM against others with only one of {RGB/Depth images} along with robot's proprioception. We report the mean, min, max success rates averaged over 3 seeds. The sensor is of the format (RGB Color (C)/Depth (D) - Third Person (TP)/First Person (FP)) as marked below the numbers. ■ indicates the mean and ● indicates the max/min values.

supplementary and *Hard* in Figure 3 *(right)*. For these experiments, we consider the following baselines:

– *TD-MPC2*: As an upper bound, we consider a TD-MPC2 trained policy with all modalities for each task and report performance without any sensor failure in the table.
– *Informed TD-MPC2*: Inspired by InformedDreamer [11], we augment TD-MPC2 to reconstruct missing modality observations. Specifically, we have decoders for each modalities that take in the latent $z$ to decode back the missing observations. The decoders are jointly trained using mean-squared error loss. To be consistent with RME, we apply modality dropout during training.
– *RME w/o Modality Dropout*: We also consider using our RME encoder without applying modality dropout – training with all modalities and then simply removing failed sensors at inference time.

**Results.** Figure 3 *(right)* shows results for the *Hard* scenario of only one high dimensional modality (along with proprioception). Note that (sensor) annotations for max and min refer to the retained sensor. In all tasks, RME *significantly* outperforms the Informed TD-MPC2 baseline (nearly 2x mean success rates across tasks) and the ablation without modality dropout. We find that RME retains approximately 70-80% of the policy performance without sensor failure on average. For pick & place tasks (first three rows), we observe that RME w/o Modality Dropout performs uniformly poorly – resulting in policies that rely on cross-modal information that generalize poorly to sensor failure. For Informed TD-MPC2 and RME, we find consistent trends that retaining third-person cameras leads to stronger performance than first-person. We provide detailed results for each modality in the supplementary.

## 5.3 Sensor failure in Sim2Real policy transfer

**Experimental setting.** We show Sim2Real transfer of a RME based policy on Cassie in blind locomotion. We test the robustness of our policy against failure of four joint encoders {left shin, right shin, left tarsus, right tarsus}. Each of these sensors measures the angular position and velocity of their respective joint links. We train a robust model-free RME policy with PPO [33] and RME (denoted as RME-PPO), where we use proprioceptive masking as described above (Fig. 2). The RME takes in a 38-dimensional proprioceptive state as input and the policy outputs a 10-dim action. We train the policy with domain randomization [51] and single contact reward as proposed in [12]. We train the policy in two stages of 500M steps each. In the first stage, we train with dynamics randomization and in the second stage we introduce random external forces to train a robust policy for Sim2Real transfer. Please find more details in the supplementary.

**Evaluation Protocol**: For evaluation of the policy in the real world, we test the commandability of Cassie which measures how well the policy follows a given the user command. Concretely, we evaluate on two metrics – commandability in linear direction (linear) which measures how much time it take for Cassie to cover a distance of 8 feet (2.43 meters) at 0.25 m/sec speed, and

| Sensor failed | Commandability x-direction (sec) | | Commandability Angular drift (sec) | |
|---|---|---|---|---|
| | Base Policy (No Sensor Failure) | RME-PPO (Ours) | Base Policy (No Sensor Failure) | RME-PPO (Ours) |
| None | 7.96 ±0.08 | 10.07 ±1.41 | 25.04 ±0.46 | 25.1 ±0.51 |
| left-shin | ✗ | 9.25 ±1.35 | ✗ | 25.21 ±0.48 |
| left-tarsus | 12 ±2.43 | 7.75 ±0.61 | 30.00 ±1.24 | 25.80 ±0.65 |
| right-shin | ✗ | 9.21 ±1.4 | ✗ | 25.41 ±0.64 |
| right-tarsus | 13 ±1.40 | 8.11 ±0.12 | 32.46 ±0.51 | 26.85 ±0.35 |
| both-shin | ✗ | 8.73 ±1.75 | ✗ | 25.44 ±0.73 |
| all 4 senors | ✗ | 7.65 ±0.04 | ✗ | 25.37 ±0.46 |

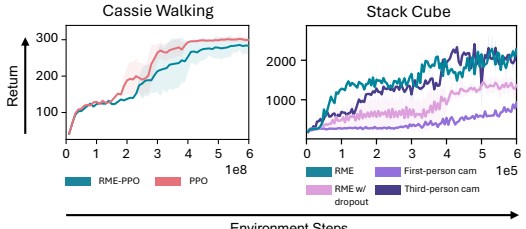

Table 1: *Commandability of Cassie*: We evaluate how well Cassie responds to input linear velocity (x-direction) and angular velocity (turn-rate). Values averaged over 2 trials for each entry of the table.

Figure 4: *Sample-Efficiency of RME* We observe either comparable (bipedal locomotion) or even increase in sample-efficiency when compared to certain modalities (eg. Stack Cube; third-person cam). Shaded areas are 95% bootstrapped CIs over 3 seeds.

commandability in angular direction (angular) that measures how much time it takes Cassie to rotate $360°$ (at a fixed position) at 15 degrees/sec. For a well trained policy, the time for linear travel should be around 9.72 sec and time for rotation should be around 24 sec.

**Results.** Based on the results in Table 1, we observe that in most instances where a sensor has failed, the original base policy with fixed value imputation of 0, fails to walk and topples. In contrast, RME not only consistently walks without falling but the linear and angular commandability is close to the optimal time even when *all* the four sensors fail, showcasing the robustness that RME provides. We provide additional videos in the supplementary material.

## 6 Discussion and Limitations

**Sample-efficiency, wall-clock time, and trainable parameters of RME:** The additional parameters that RME adds on top of TD-MPC2 are those for the multimodal transformer (2-layer, 4-head) – an increase from a base 5M trainable parameters to 7.2M. Averaging over all RME+TD-MPC2 models presented in this work, adding RME increases wall-clock runtime of training by 72%; however, Walker skews this substantially and the average drops to 27% if it is excluded. In terms of sample efficiency, we find little difference in the model-free experiment on Cassie, (Fig. 4). While in the case of robotic manipulation, we observe that the sample-efficiency of RME-TDMPC2 is either comparable (third-person cam) or even better than individual modalities (first-person-cam) for StackCube. We note that having all modalities in the case of StackCube is not as sample-efficient as RME, potentially due to state over-specification problem [55]. These increases in wall-clock time and reductions in sample-efficiency are a trade-off for significantly increased robustness to missing modalities.

**Limitations**: A key assumption of our work is that the agent knows which sensors have failed or are corrupted, which in itself is a challenging problem [56] that is beyond the scope of this work. Another limitation of our work is that the proposed dropout mechanism in RME assumes that the agent can correctly predict actions with the information from remaining modalities after dropout during training – which may not always be true and could cause difficulties in policy learning.

## 7 Conclusion

In this work, we investigate the effectiveness of RL agents in the presence of sensor failure during robot manipulation and locomotion tasks. Our findings indicate that RL policies trained with a standard multimodal encoder do not generalize well to various sensor failure scenarios. To address this issue, we propose RME, a multimodal encoder design incorporating additional information about missing modalities as input and a training strategy based on modality dropout. Our encoder design is compatible with both low- and high-dimensional input modalities. Experimental results, conducted in both simulation and on a real-world bipedal robot Cassie, demonstrate that RL policies trained with the proposed encoder and training strategy maintain effectiveness even under severe sensor failure scenarios, significantly outperforming the baseline RL policies.

**Acknowledgments**

The authors would like to thank the anonymous reviewers for their insightful feedback that helped improve the quality of the paper. SP would like to thank Mohitvishnu Gadde and Pranay Dugar with their help conducting real world experiments on Cassie and DMV & ViRL labmates for their feedback on the draft. This work is supported in part by NSF Award No. 2321851 and DARPA TIAMAT Agreement No. HR0011-24-9-0423. The views and conclusions contained herein are those of the authors and should not be interpreted as necessarily representing the official policies or endorsements, either expressed or implied, of the U.S. Government, or any sponsor.

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

# 8 Appendix

## 8.1 Additional Related Works

**Handlind Missing modalities.** Another work [57] trains hierarchical RNNs, where the low level RNN handles individual modalities and the higher level RNN handles the fused representation from the individual modality representations. Although their approach, in theory, can work with more than 2 modalities, they restrict all their experiments to two modalities, and it is non-trivial whether the extension to multiple modalities would work well with a large number of sensors because multiple modality representations are fused via concatenation in an RNN, similarly as in an MLP network used in [23].

**Large Multimodal policies**: Octo [58] and RT-X [59] models train a large multi-modal policies that are capable of handling multiple robotic embodiments, action spaces. These large transformer based models are trained using the causal masking. However, all such transformer based models are limited to imitation learning where access to high fidelity video demonstrations are assumed.

**Model based RL (MBRL).** A sample-efficient way to train learnable control policies is via model-based RL, which learns the model of the environment [60] and uses it for policy learning [61, 62]. In the case of high-dimensional modalities such as images, a popular approach is to learn the environment dynamics in a compact latent space that is supervised using rewards [63, 32] and image reconstruction [64, 65, 34, 66]. Recently, Scaffolder [20] proposed learning two dynamics models—one for privileged inputs and another for the target modalities—and distilling representations from the former. Similarly, TWIST [67] also proposed distilling the teacher world model to the student world model as a means for efficient Sim2Real transfer. For multiple modalities, both these methods require the number of dynamics models to increase linearly with the number of modalities, in contrast to our work, which aims to learn a compact abstraction from various modalities and develop a single dynamics model. More importantly, unlike our method, these models would not generalize to situations where some of the input modalities fail.

## 8.2 Details of robotic environments

We consider 5 tasks (four from DM Control suite [50] and one task on Cassie-walking [51]) for the low dimensional proprioceptive sensory failure, 5 tasks for high-dimensional modality failures from the ManiSkill2 [54] suite as shown in Figure 5. The dimensionality of the observation space and action space is mentioned in Table 2.

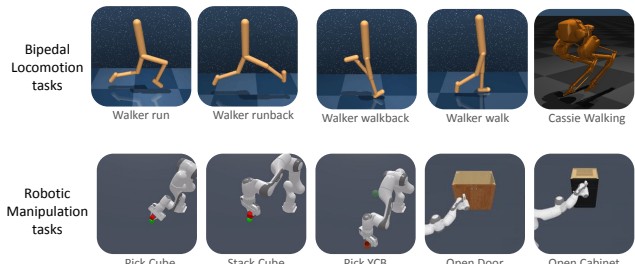

Figure 5: *Tasks* – We consider 5 bi-pedal locomotion tasks (top) and 5 robotic manipulation tasks (bottom). We test the low-dimensional sensory failures (e.g., the robot's proprioception) in the locomotion task suite and the high-dimensional sensory failures (e.g., cameras) in the manipulation task suite.

## 8.3 Details on Cassie bipedal walking experiment

Setting: We consider sensory failures in 4 joint links of Cassie, a bipedal robot. Specifically, we simulate the failure in `shin` and `tarsus` links on both the legs. We train the policy following the reward function design in [12] and train the model in 2 stages. The first stage considers dynamics and environment state variable randomization, and the second stage involves learning a robust policy

| Task | Observation dim | Observation type | Action dim |
|------|-----------------|------------------|------------|
| Walker-Walk | 24 | Proprioception | 6 |
| Walker-Run | 24 | Proprioception | 6 |
| Walker-WalkBack | 24 | Proprioception | 6 |
| Walker-RunBack | 24 | Proprioception | 6 |
| Cassie-Walk | 39 | Proprioception | 10 |
| Pick Cube | $(64 \times 64 \times 3)$ | RGB & Depth (TP, FP) | 4 |
| Pick YCB | $(64 \times 64 \times 3)$ | RGB & Depth (TP, FP) | 7 |
| Stack Cube | $(64 \times 64 \times 3)$ | RGB & Depth (TP, FP) | 4 |
| Open Door | $(128 \times 128 \times 3)$ | RGB & Depth (TP, FP) | 11 |
| Open Drawer | $(128 \times 128 \times 3)$ | RGB & Depth (TP, FP) | 11 |

Table 2: State and Action spaces of each task in the locomotion and manipulation suite.

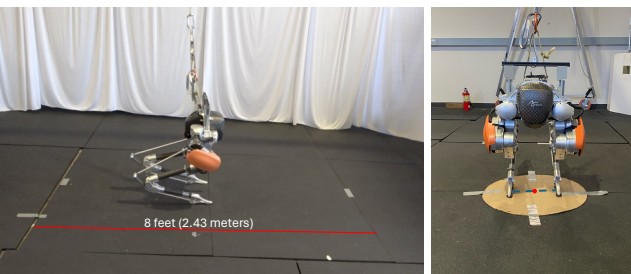

Figure 6: *Measuring Commandability of locomotion policy on Cassie* (left) commadability in the `x-direction` and (right) `angular drift`

under random external forces (of 30N) with a probability of 0.1 acting on Cassie. We follow this training procedure from [12]. Each of the stages is trained for 500M steps each.

Metrics: It is important to measure a set of quantifiable metrics to measure the progress in real-world robot deployments. For this we adopt two metrics from [12] that measure the *commandability* of the robot in `x-direction` and `angular drift`. The evaluation setup is shown in Figure 6.

*Commandability in x-direction.* We consider a distance of 8 feet (or 2.43 meters) as shown in Figure 6 and command Cassie to travel at a speed of 0.25 m/s. For a near-optimal policy, it takes about 9.72 seconds. We then measure how long it takes for an RME based policy to travel the distance and report it in Table 1.

*Commandability to measure angular drift.* We consider a circular region with a diameter of 50cm and initialize Cassie at the center of the circle (center of Cassie is defined as the mid-point of line joining the feet of Cassie as shown by the ● in Figure 6). We command an angular speed of 15 degree/s and measure the time taken to complete a single rotation ($360°$). We consider the trial successful if the center of Cassie is withing 10 cm from the circumference of the circle. For an optimal policy it takes 24 seconds and we compare this against RME based policy with sensory failures to see how much additional time it takes for our policy (Table 1).

## 8.4 Architecture Details

RME **multimodal transformer**: For the `base` TD-MPC2 model (5M parameters), the multimodal transformer consists of 2 encoder blocks – each consisting of multi-head attention (MHA) and feedforward network [42]. We consider 16 heads for MHA module. The input to the transformer is the encodings from each modality and the `[CLS]` token which are all of $512$ dimensions each.We also generate fixed 512 dimensional modality type embeddings and add them to the corresponding modality encodings. The hidden dimension of the feedforward network in each encoder block of transformer is 256.

The output of the RME is the output of the attention layer on the [CLS] token attending to the other input tokens of the transformer, which produces a 512-dimensional latent vector. This is the input to the policy. The policy network for TD-MPC2 is implemented as a 3-layer MLP network with SiLU activation, and for model-free PPO training of Cassie it is implemented as a 2-layer LSTM[68] network.

**Imputation Network**: For the locomotion tasks, we consider a 3 layer MLP with LayerNorm [52] and Mish activation function [53] with the final output layer as linear layer. We train one imputation network for predicting each proprioceptive sensor value for 200 epochs with a learning rate chosen from {1e-5, 5e-5, 1e-4}. The network parameters are optimized by Adam [69] and the transitions from the replay buffer of RME are used to train the network with a 80-10-10 (`train-val-test`) split.

**Decoder architecture for Informed TD-MPC2 baseline.** We use the decoder structure from [36]. We show the architecture in Table 3.

| Layer | Channels | Kernel Size | Norm/Activation |
|---|---|---|---|
| ConvTranspose2d | 512 | 2 | LayerNorm/SiLU() |
| ConvTranspose2d | 256 | 2 | LayerNorm/SiLU() |
| ConvTranspose2d | 128 | 4 | LayerNorm/SiLU() |
| ConvTranspose2d | 32 | 4 | LayerNorm/SiLU() |
| ConvTranspose2d | 3 | 2 | LayerNorm/SiLU() |
| Sigmoid() | - | - | - |

Table 3: Decoder Architecture for Informer-TDMPC2 baseline.

## 8.5 Additional results for sensor failures in high-dimensional modalities (`Easy`)

In this section, we present the results for sensory failures in high-dimensional modalities where only one of the modalities is dropped during evaluation. Specifically, for all tasks either the RGB or the depth modalities from the first-person (FP) or the third-person (TP) cameras are dropped out. Since only one of the modality is dropped, we denote this setting by `Easy` as compared to only operating on a *single* modality (`Hard`) as shown in Table ?? of the main paper.

| Task | Base Policy (No Sensor Failure) | Informed TD-MPC2 | | | RME (w/o Modality Dropout) | | | RME (Ours) | | |
|---|---|---|---|---|---|---|---|---|---|---|
| | | max (sensor) | min (sensor) | mean | max (sensor) | min (sensor) | mean | max (sensor) | min (sensor) | mean |
| PickCube | 1.0 ±0 | 0.79 ±0.17 (C-FP) | 0.69 ±0.15 (D-FP) | 0.72 ±0.05 | 0.30 ±0.10 (C-FP) | 0.10 ±0.03 (C-TP) | 0.17 ±0.11 | 0.91 ±0.06 (D-FP) | 0.80 ±0.20 (D-TP) | 0.86 ±0.05 |
| PickYCB | 0.83 ±0.30 | 0.65 ±0.08 (C-FP) | 0.60 ±0.19 (C-TP) | 0.62 ±0.02 | 0.40 ±0.11 (D-TP) | 0.07 ±0.01 (C-TP) | 0.18 ±0.14 | 0.75 ±0.01 (D-FP) | 0.81 ±0.06 (D-FP) | 0.79 ±0.04 |
| StackCube | 0.96 ±0.05 | 0.74 ±0.13 (C-FP) | 0.51 ±0.21 (D-FP) | 0.72 ±0.10 | 0.31 ±0.18 (C-FP) | 0.02 ±0.03 (D-TP) | 0.15 ±0.14 | 0.89 ±0.09 (C-FP) | 0.71 ±0.20 (C-TP) | 0.82 ±0.07 |
| OpenDrawer | 0.81 ±0.20 | 0.73 ±0.11 (C-FP) | 0.69 ±0.18 (C-TP) | 0.70 ±0.01 | 0.27 ±0.05 (D-TP) | 0.09 ±0.01 (D-FP) | 0.19 ±0.09 | 0.80 ±0.11 (D-TP) | 0.77 ±0.08 (C-FP) | 0.79 ±0.01 |
| OpenDoor | 0.76 ±0.30 | 0.66 ±0.10 (C-TP&D-FP) | 0.61 ±0.05 (C-FP) | 0.65 ±0.02 | 0.40 ±0.08 (C-TP) | 0.11 ±0.27 (D-FP) | 0.28 ±0.13 | 0.81 ±0.23 (D-FP) | 0.73 ±0.08 (C-TP) | 0.77 ±0.04 |

Table 4: *Robustness towards modality dropout during deployment* (`Easy`). We test MMWM against others *by removing one of* {RGB/Depth images}. We report the mean, min, max success rates averaged over 3 seeds. The sensor is of the format (RGB Color (C)/Depth (D) - Third Person (TP)/First Person (FP)) as marked below the numbers.

**Results:** We show the results for the `Easy` scenario in Table 4. Not surprisingly, we observe that removing only one modality during evaluation leads to better performance across all baselines. However, there still remains a considerable gap between Informed TD-MPC2 and RME. Additionally, for the case of RME w/o Modality Dropout, we observe that even removal of a single modality can have huge impacts with a relative **357%** drop in performance compared to RME. For the `Easy` scenario, the RME has a relative increase of 18.1% compared to Informed TD-MPC2 baseline.

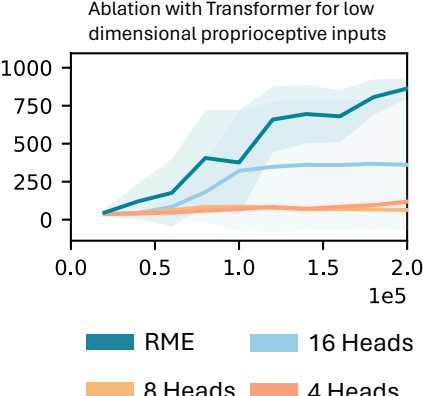

Figure 7: *Ablation study on instability of using low-dimensional sensors as direct input to transformer*

## 8.6 Wall-clock time

In Table 5, we present the wall-clock running time on a single NVIDIA Quadro RTX 8000 GPU. The increase in relative wall-clock time because of addition of RME is 27% when we don't include the Walker tasks. Adding the walker tasks skews the results to a relative 72% increase in the wall-clock time. However, the Walker tasks take the least amount of time to learn (3 hours), so the impact of such an increase is not significant in practice.

| Task | TD-MPC2 ↓ | RME ↓ |
|---|---|---|
| Walker | 1 | 3.1 |
| Cassie | 70 | 70.1 |
| PickCube | 13 | 20 |
| StackCube | 25 | 32 |

Table 5: **Wall-clock training time in (hrs).**

## 8.7 Ablation: Stability of Transformer as opposed to Masking MLP for low-dimensional proprioceptive inputs

As mentioned in Section 4, we observed that, to accommodate low-dimensional sensors such as proprioception, treating each low-dim sensor as an independent input after encoding via MLP led to instability in training. We show one such result in Table 7 on Walker Walk, where only 1 seed for a transformer with 16 heads in the multi-head attention led to a successful policy. Remaining 4 seeds as well as *all* seeds for 8 and 4-head transformer layer led to no successful training. TThis could be potentially due to the difficulty of converting a small-dimensional vector, such as the one in proprioceptive input, into a meaningful 512 dimensional modality encoding vector via MLP. This also can add redundant dimensions in the latent space, slowing down the transformer's training

## 8.8 Compounding errors in imputation

We additionally analyze how the error of the imputation network behaves as a function of time steps. Specifically, for the imputation baseline in Cassie simulation, we plot the mean-squared error between the predicted missing sensor values and the ground truth value. The imputation network is trained to predict the current timestep's value based on a history of the previous 10 timesteps.
**Observation:** We observe that the initial 5-10 timestep predictions of the imputation network are accurate. However, the prediction quickly worsens around t=15-20. Visually, this corresponds to Cassie finding it hard to balance its legs on the ground and after this point, it tries to maintain its balance before falling onto the ground in about 250 timesteps. We hypothesize that states that correspond to this peak in the plot are potentially underexplored in the replay buffer on which the

imputation network was trained – hence causing a compounding effect in the imputation prediction values, and thus causing the robot to fail.

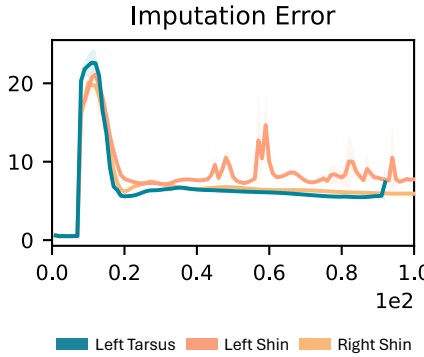

Figure 8: *Imputation error as a function of time steps (on horizontal axis).*

## 8.9 Network dropout

We performed three additional experiments to evaluate the robustness of network dropout towards missing modalities. For this, we experiment with Dropout [70] for low-dimensional modalities and for high dimensional modalities we perform SpatialDropout [71] as well as DropBlock[72]. During inference, for all the experiments, dropout is turned off. We show the results for these experiments in Figure 9.

**Network dropout for low-dimensional sensors**: We perform experiments with dropout rates of 0.2 and 0.5. With a dropout rate of 0.2, we find that the robustness of the policy to missing sensors did increase compared to the base policy with zero-imputation (Figure **??**), however with a rate of 0.5, we found that the models struggled to learn a good policy. Hence, we observe a significant drop in the performance as shown in Figure 9 (left). We hypothesize that 0.5 is too high of a dropout rate for the encoder to obtain a meaningful representation to the policy.

**Network dropout for high-dimensional modalities**: We perform two kinds of network dropout for the CNN layers – SpatialDropout [71] which drops entire features by zeroing out them and DropBlock [72] that drops a contiguous block of features. While in general, we find that BlockDrop performs slightly better than SpatialDropout, performance of both is still lags far behing RME as shown in Figure 9 (right).

## 8.10 Including imputation baselines during training

In Section 5.1 we presented results for imputation where the imputation model was trained post-policy training using the replay buffer collected by the RL algorithm during training. In this section, we present two additional results on Walker environment where the (a) zeros-imputation and (b) learned-imputation models are part of the policy training loop.

Specifically, for (a), we perform zero-imputation and feed the representation obtained by the encoder to the policy and for (b), we trained the imputation network simultaneously and used the predictions of the imputation network to feed to the policy. We show the results of this experiment in Figure 10. We observe that while compared to post hoc accommodation, imputation models trained in the RL policy training improves the robustness of the policy to sensor failures, it still lags significantly behind an RME type architecture.

One important thing to note is that the *maximum return* for all the baselines, especially for Walk and Walk backwards is on par or even better than RME and we consistently notice that this is because of the Right Foot sensor which when dropped in testing leads to a decent policy –making us infer that the right foot sensor for the simulated Walker task is potentially redundant.

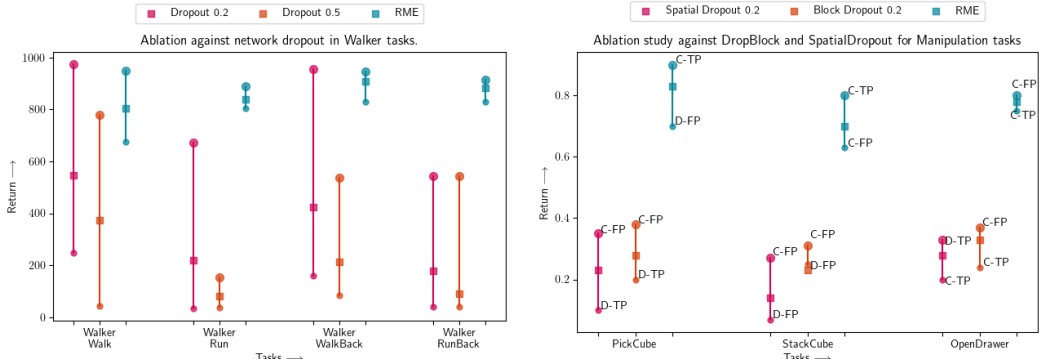

Figure 9: *Comparison against network dropout*: We perform Dropout[70] for encoder networks in Walker tasks and SpatialDropout [71] and DropBlock[72] for manipulation tasks and observe consistently that all these network dropout strategies lead to a sub-optimal performance when compared to RME. ■ indicates the mean and ● indicates the max/min values.

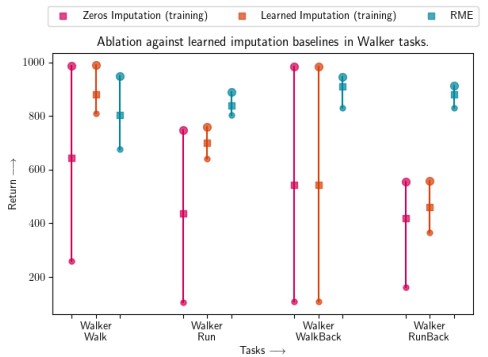

Figure 10: *Imputation during training:* We perform both zeros-imputation and learned imputation as a part of the RL policy training and observe that while the performance of including them in training loop is better than imputation post policy training, the imputation methods still lag behind in their performance compared to RME. ■ indicates the mean and ● indicates the max/min values.

Figure 11: *Ablation of RME without transformer* We observe an average improvement of 93% across three robotic manipulation tasks of RME when compared to a policy that is trained without transformer encoder. ■ indicates the mean and ● indicates the max/min values.

## 8.11   Ablation without RME with sensor dropout

Another experiment we performed was to test the effectiveness of the proposed RME' transformer structure. Instead of using a transformer as done in RME, we used an MLP network with modality feature masking to indicate missing modalities as an encoder, similar to [23]. We present our results for robotic manipulation experiments in Figure 11. As expected, this is clearly better than training RME without any modality dropout during training; however, we find that this is still less performant compared to the proposed RME encoder+dropout (with an average increase of 93% success rate), indicating the benefit of using a transformer based architecture.

## 8.12   Effect of history and memory in imputation baseline

In our imputation baselines, our imputation network takes in past 10 timesteps of the state of the robot and predicts the next state. The imputation network was trained by *randomly sampling* 10 consecutive states from the replay buffer and using that to predict the next (11th) state. During the inference, we have access to the history of ground truth state information that we get from the environment (except for the masked sensor) which is the input to the imputation network. We

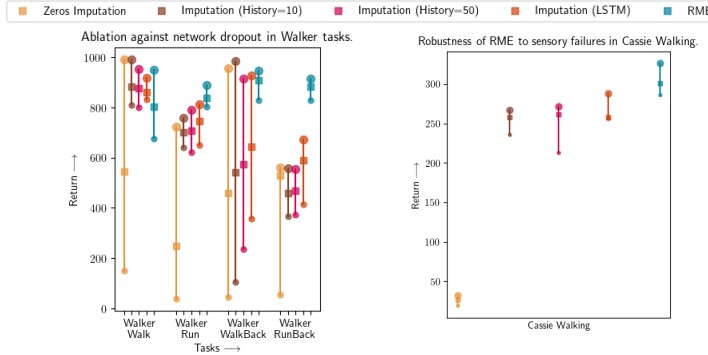

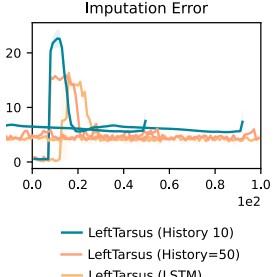

Figure 12: *Effect of history length and memory for imputation model:* We compare RME against different models of imputation network with different lengths of history (10, 50) as well as an LSTM network. We do not observe significant differences between different learned imputation baselines, albeit the average performance of the LSTM network is slightly better. ■ indicates the mean and ● indicates the max/min values.

Figure 13: *Imputation error as a function of time steps (on horizontal axis).* We find that the LSTM architecture for imputation network does incur a high error slightly later than an MLP with 10/50 timesteps history.

further experiment with (a) higher history length of 50 and (b) an LSTM [68] architecture to see if this extended history leads to better robustness under sensor failures. As we showed in Section 8.10 that including the imputation in the RL policy loop performs better than post hoc imputation, we perform these experiments under that configuration.

As shown in Figure 12, we find no significant difference between different levels of history for imputation networks. The LSTM architecture for the imputation model performs slightly better on average in terms of returns, however its overall performance is still behind RME. As mentioned in Section 5.1, we hypothesize that this may be because the errors in learned imputation may have compounding effects if predicted sensor values drive the policy to out-of-distribution states where the imputation network is less well trained.

Further, we also extend the imputation error analysis on the new imputation models (History=50 steps and LSTM) as shown in Figure 13. We find that the LSTM model does incur a high error albeit at a slightly later timestep than the MLP model trained on history of 10 or 50 steps history.

