# OpenReview forum: "Simple Masked Training Strategies Yield Control Policies That Are Robust to Sensor Failure"
_robot-learning.org/CoRL/2024/Conference — CoRL 2024_

### Official Review · Reviewer_SZc2 · 2024-07-13
**Relevant problem tackled well but suffers from critical assumption**

**Originality:** 3
**Technical Quality:** 4
**Clarity Of Presentation:** 3
**Potential Impact:** 3
**Recommendation:** 4
**Confidence:** 4

**Review:**

The paper tackles a relevant problem in its aim to foster robustness in robotic behavior. Missing sensor data is a realistic problem and requires suitable solutions. The paper is generally well written and the information provided in the paper is clearly accessible. The evaluation useful, as is the video and supplementary. It is great that the method is evaluated for locomotion as well as manipulation tasks.
However, the paper also has weaknesses.

(1) The biggest weakness in my opinion is that the paper builds on a weak / erroneous assumption. This assumption is that the robot's actions would not be influenced if sensors fail. However, this is most likely not true. The authors test the Cassie robot in cases when proprioceptive sensors fail, but if an encoder fails on a brushless motor (I believe cassie uses BLDCs) then it cannot be controlled in a closed loop manner anymore. Open loop torque or velocity control might be possible, but the authors do not state that either of these types of control is used and I doubt this is the case as it would be highly ineffective. The authors do not state what the action space in either experiment looks like and thus we cannot assess if the the robot's control would remain unimpeded, and thus we cannot assess if the method would actually work with a failed sensor (which is the claimed contribution of the method).

(2) There is very relevant related work that the paper does not refer to and that it thus does not contrast and compare itself against. Two such works are:

H. Ichiwara, H. Ito, K. Yamamoto, H. Mori and T. Ogata, "Modality Attention for Prediction-Based Robot Motion Generation: Improving Interpretability and Robustness of Using Multi-Modality," in IEEE Robotics and Automation Letters, vol. 8, no. 12, pp. 8271-8278, Dec. 2023

Lee, Michelle A., Matthew Tan, Yuke Zhu, and Jeannette Bohg. "Detect, reject, correct: Crossmodal compensation of corrupted sensors." In 2021 IEEE international conference on robotics and automation (ICRA), pp. 909-916. IEEE, 2021.

(3) The paper does not provide clear information about the shape of the full policy. We received information about the encoder/transformer part. But it's unclear how exactly the output of the transformer maps to robot actions.

(4) Line 199 -- Somehow ambiguous, but if prior work refers to the authors themselves this may be a violation against double blind... "as in prior work [56]."

(5) Minor typos, see Questions for Rebuttal

**Quality Of The Limitations Section:**

3

**Questions For Rebuttal:**

- The authors need to clarify what the action space is in each experiment and deliver a realistic explanation of why the actions are still executable with failed sensors. If this is not possible, then the method needs to tune down its claim* or adapt the method such that it can cope with impeded action capabilities. (*the contribution is more theoretical than practical in this case. the method does not actually yield behavior robust against corrupted sensors, but against cases where sensor information is available to low-level control but not somehow does not arrive at the policy level)

- The related work should refer to the two works I mentioned above, but ideally also search for other works of related similarity level.

- The paper should cleary and explicitly provide information how the RME is part of the policy and what the full policy looks like.

- Clarify what is sg() in equation 2.

- 129 - typo - Intergral

- 164 - typo - multi=

- 179 grammar? concatenated to the vector of observations

**Robotics Focus:**

4

**Summary Of Paper:**

The paper presents a method to learn robot policies that are robust against sensor failure. During training sensor failure is simulated using modality dropout and a transformer-based module called Robust Multimodal Encoder (RME) learns to attend to different input modalities to robustly create a latent representation. The method is evaluated in locomotion and evaluation tasks and the results show that the method is more robust against missing sensor modalities than comparable baselines.

**Summary Of Recommendation:**

It's a great paper that proposes a promising method for generating behavior robust to sensor failure.

---

### Official Review · Reviewer_X6ey · 2024-07-18

**Originality:** 2
**Technical Quality:** 3
**Clarity Of Presentation:** 4
**Potential Impact:** 3
**Recommendation:** 4
**Confidence:** 3

**Review:**

## Strengths
- This paper addresses a key problem faced by practicing roboticists, and it proposes an easy-to-use set of tools for solving that problem.
- The paper is clearly written and easy to follow.
- The paper includes evaluation on multiple tasks in simulation and evaluation in HW.

## Major concerns
1. **Evaluation:** This paper seems to claim two main contributions: a transformer-based encoder for multi-modal sensor fusion (RME), and a sensor-dropout scheme for training policies that are robust to sensor failure. The experimental evaluation in Section 5.2 does a good job of evaluating these contributions separately (comparing RME+dropout with RME and a different observer + dropout), but the evaluation in Section 5.1 falls short. In particular, I have the following concerns with section 5.1:

a) The baselines seem quite weak (see my question about the performance of the learned imputation method).

b) The comparisons are not apples-to-apples. The actor policy for RME is trained simultaneously with RME, while the actor for learned imputation is trained without imputation and transferred zero-shot.

I am particularly curious as to how well the base policy (without RME) trained with sensor dropout would perform. i.e. is most of the benefit due to sensor dropout or to the transformer encoder?

2. **Novelty:** Although I think that seemingly "simple" ideas like sensor dropout can be strong contributions (if paired with strong evaluation and novel insights), but I have concerns that sensor dropout has already been explored in prior work (see [1]). If the effect of sensor dropout is already known, is the primary contribution of this paper the RME architecture for encoders? In that case, I would like to see more evaluation of how alternative encoder architectures perform when trained identically to RME.

## Minor concerns
- Some missing experimental details (e.g. in Table 1, how many trials are used to compute these mean rewards?)
- Quantitative data in tables can be hard to read (consider a plot instead, and include exact data in tables in the supplementary material if needed).

[1] : [Liu et al, "Learning End-to-end Multimodal Sensor Policies for Autonomous Navigation," CoRL 2017](https://proceedings.mlr.press/v78/liu17a.html)

**Quality Of The Limitations Section:**

3

**Questions For Rebuttal:**

- For the learned imputation baseline: is the policy trained with the learned imputation mechanism present? And if so, is the learned imputation network trained or fine-tuned simultaneously with the policy network?
- Similarly for the zero imputation baseline, is the control policy trained with the zero-imputation mechanism and sensor failures?
- Would Table 1 be better presented as a plot? e.g. a scatterplot or boxplot to show the distribution of returns. Similar for Table 2.
- Why do the results for Walker skew the wall-clock time increase?
- Fig 7 in the supplementary materials: It's not surprising that the imputation network fails after 10 timesteps if it only has 10 timesteps of memory. Would you see the same behavior from an imputation network that has memory or estimates some sort of latent state? Fig. 7 makes me think that the baseline you have chosen for the imputation network is not particularly strong. Did you recreate any of the sensor imputation baselines mentioned in the related work section?

**Robotics Focus:**

4

**Summary Of Paper:**

The paper proposes two strategies for improving the robustness of learned robot control policies to sensor failures. The first is a training scheme that includes random sensor dropout, and the second is a multi-modal encoder architecture that is robust to sensor dropout. The paper evaluates these components on multiple tasks in simulation and hardware.

**Summary Of Recommendation:**

My concerns about the evaluation and novelty of this paper's contributions are substantial enough that I recommend rejection, but I would be open to upgrading if my concerns were addressed. [UPDATE] The authors have addressed my concerns and I am in favor of accepting the revised paper.

---

### Official Review · Reviewer_iBv4 · 2024-07-24
**Review for Submission 605**

**Originality:** 2
**Technical Quality:** 3
**Clarity Of Presentation:** 3
**Potential Impact:** 2
**Recommendation:** 3
**Confidence:** 3

**Review:**

Strengths:
- The paper has clear figures and is overall well written.
- The paper provides comparison with alternative approaches such as imputation in their experimental evaluation.
- The paper shows compatibility with both visual and non-visual sensors for their architecture design.
- The paper shows efficacy of the approach with sim2real.

Weaknesses:
- Some concerns about novelty. Approaches like Octo [1] has shown that that causally masking out modalities can enable flexible sensor modalities for robotic manipulation. In this work they consider including other visual features (like wrist and 3rd person cameras) as well as proprioception inputs.
- Other forms of sensor failure not considered. Sometimes sensor failures are more subtle than the sensor not providing a reading (e.g camera miscalibration or noisy sensors).

References:
1. Octo Model Team, et al. “Octo: An Open-Source Generalist Robot Policy.” Proceedings of Robotics: Science and Systems, 2024, Delft, Netherlands. https://arxiv.org/pdf/2405.12213

**Quality Of The Limitations Section:**

3

**Questions For Rebuttal:**

1. How do other alternate masking approaches perform (e.g adding dropout in the decoder/encoder/both, blockwise) compare to modality dropout? Are these approaches also able to recover the performance?
2. Are there sensor failures in the training dataset or are these filtered out? If they are filtered, wouldn't this prevent modality dropout from naturally arising?
3. Does a pre-trained encoder ameliorate some of this issues if for example they were trained with some masking (e.g MAE/VIT with patch dropout)?

**Robotics Focus:**

4

**Summary Of Paper:**

Provide a masking strategy to mitigate sensor failures affecting robotic control.

**Summary Of Recommendation:**

Well written paper studying the affect of masking on sensor failures. However, have some concerns about novelty given prior work.

---

### Author Rebuttal · Authors · 2024-08-14

Dear Reviewers and Area Chair,

We are uploading a revised manuscript with all the changes we have made in red (in a diff-format).
- We have a dedicated section for the Rebuttal (Sec. 8).
- We have replaced the tables with scatter plots for better readability.
- We have added clarifications on action space as well as the policy structure.
- We have expanded the related works section after the reviewers pointed us to more related works.

---

### Decision · Program_Chairs · 2024-09-04

**Decision:**

Accept

**Comment:**

The reviewers and area chair have identified the following strength and weaknesses with the initial submission:

Strengths:
- the problem the paper addresses is very relevant
- the paper is well-written and easy to follow
- the paper provides useful experiments in simulation and the real world on both locomotion and manipulation

Weaknesses:
- Novelty: the idea of drop-out has been explored in prior work and the authors should revisit their novelty claims and related work section
- Scope & assumptions: the authors only consider complete sensor breakdown’s while more subtle sensor failures are possible; the assumption that actions are not affected by sensor failure seems strong.
- Baselines: The authors should revisit their baselines to ensure that the comparisons to the proposed approach are fair and support the claims

Post-Rebuttal, the authors have made a substantial effort to clarify their contribution and added multiple baselines as required by the reviewers. The results are compelling leading to an accept.